# Mouse Vendor Influence on the Bacterial and Viral Gut Composition Exceeds the Effect of Diet

**DOI:** 10.3390/v11050435

**Published:** 2019-05-13

**Authors:** Torben Sølbeck Rasmussen, Liv de Vries, Witold Kot, Lars Hestbjerg Hansen, Josué L. Castro-Mejía, Finn Kvist Vogensen, Axel Kornerup Hansen, Dennis Sandris Nielsen

**Affiliations:** 1Department of Food Science, Faculty of Science, University of Copenhagen, 1958 Frederiksberg, Denmark; devries.liv@gmail.com (L.d.V.); jcame@food.ku.dk (J.L.C.-M.); fkv@food.ku.dk (F.K.V.); 2Department of Veterinary and Animal Sciences, Faculty of Health and Medical Sciences, University of Copenhagen, 1870 Frederiksberg, Denmark; akh@sund.ku.dk; 3Department of Environmental Science, Aarhus University, 4000 Roskilde, Denmark; wk@envs.au.dk (W.K.); lhha@envs.au.dk (L.H.H.)

**Keywords:** Bacteriophages, gut microbiota, animal model reproducibility, vendor effect, virome

## Abstract

Often physiological studies using mice from one vendor show different outcome when being reproduced using mice from another vendor. These divergent phenotypes between similar mouse strains from different vendors have been assigned to differences in the gut microbiome. During recent years, evidence has mounted that the gut viral community plays a key role in shaping the gut microbiome and may thus also influence mouse phenotype. However, to date inter-vendor variation in the murine gut virome has not been studied. Using a metavirome approach, combined with 16S rRNA gene sequencing, we here compare the composition of the viral and bacterial gut community of C57BL/6N mice from three different vendors exposed to either a chow-based low-fat diet or high-fat diet. Interestingly, both the bacterial and the viral component of the gut community differed significantly between vendors. The different diets also strongly influenced both the viral and bacterial gut community, but surprisingly the effect of vendor exceeded the effect of diet. In conclusion, the vendor effect is substantial not only on the gut bacterial community but also strongly influences viral community composition. Given the effect of GM on mice phenotype, this is essential to consider for increasing reproducibility of mouse studies.

## 1. Introduction

During the last decade, the gut microbiome (GM) and its role in host health and disease have emerged as a rapidly expanding area of research [1,2]. Most GM studies focus on the bacterial gut component, whereas the archaeal, yeast, fungal, and viral (virome) components of the GM have been more sparsely investigated [3,4]. However, recently gut virome dysbiosis has been associated with flares of Crohn’s disease and ulcerative colitis [5], *Clostridium difficile* associated diarrhea [6] and type-2-diabetes [7] highlighting the importance of the virome in health and disease. The gut virome is predominated by prokaryotic viruses [8], also called bacteriophages (phages) which are viruses attacking bacteria in a host-specific manner and is estimated to exist at least in the ratio of 1:1 to bacteria in the gut [9]. Although more in-depth studies are needed, phages are hypothesized to play an important role in shaping the bacterial GM component [3,10,11]. This has been further supported by Draper et al. that applied fecal virome transplantation (FVT) to reshape the mouse gut microbiome after antibiotic treatment [12], and Ott et al. that treated *C. difficile* patients with FVT and obtained a similar efficacy as with fecal microbiota transplantation (FMT) [13]. Interestingly, it has been shown that the bacterial and virome component of the GM respond to perturbations caused by a diet intervention in a desynchronized manner highlighting the potentially unique role of the virome in gut health [3]. Inbred mice strains and controlled environments are often applied to minimize inter-individual variation. However, it has recently been questioned whether inbred mice really have a lower inter-individual variation compared to outbred mice [14], and traits previously thought to be caused by genetics have been shown to be related to the GM composition [15]. Several studies have shown that the bacterial component of the GM differs between the same mice strains obtained from different vendors [16], which directly influences the mice phenotype (e.g., disease expression) [17,18,19]. As an example, segmented filamentous bacteria (SFB) induces a robust T-helper cell type 17 (Th17) population in the small intestine of the mouse gut, but are only present in mice from some vendors [20]. Subsequently, Kriegel et al. demonstrated that SFB promotes protection against type-1-diabetes in non-obese diabetic (NOD) mice [21]. Prolonged feeding with a high-fat (HF) diet is the standard protocol for inducing an obese phenotype in mice. It is well-established, that the HF diet also changes the bacterial component of the GM [22] and that GM composition is strongly correlated to the primary readouts of this model [23]. According to Howe et al. [3] also the viral community is affected by an HF diet. Here we report how both the choice of vendor and diet will affect the bacterial and the viral composition in C57BL/6N mice purchased from three different vendors. To our knowledge, no studies have yet simultaneously examined vendor and diet-dependent effects on both the bacterial and viral GM composition in mice.

## 2. Materials and Methods

### 2.1. Animals, Diets and Tissue/Fecal Sampling

All procedures regarding the handling of the animals were carried out in accordance with the Directive 2010/63/EU and the Danish Animal Experimentation Act with the license ID: 2012-15-2934-00256. The present study included in total 54 C57BL/6N male mice purchased at age five weeks from three vendors, represented by 18 C57BL/6NTac mice (Taconic, Lille Skensved, Denmark), 18 C57BL/6NRj mice (Janvier, Le Genest-Saint Isle, France), and 18 C57BL/6NCrl mice (Charles River, Sulzfeld, Germany). Six mice from each vendor were sacrificed and sampled immediately after the arrival to assess the gut microbiome at baseline. The baseline mice were included to ensure that potential differences in gut microbial diversity and composition was not due to our housing conditions and to track if initial divergence could be maintained for 13 weeks on the respective diets. The remaining 12 mice from each vendor were upon arrival housed at ambient temperature (20–24 °C), 12 h light/dark cycle, with humidity of approx. 55%, shielded from ultrasounds >20 kHz. The mice were divided into cages of three mice and randomly organized. Cages (Cat. no. 80-1290D001, Scanbur) were enriched with bedding, cardboard housing, tunnel, nesting material, felt pad, and biting stem (respectively Cat. no. 30983, 31000, 31003, 31008, 31007, 30968 Brogaarden). One C57BL/6NTac mouse on HF diet was killed by a mouse in the same cage, and the two remaining mice were divided into individual cages. Animal housing was carried out at Section of Experimental Animal Models, University of Copenhagen, Denmark. For 13 weeks the mice were fed ad libitum high-fat diet (HF, Research Diets D12492, USA) or low-fat diet (LF, Research Diets D12450J, USA), see Figure 1. According to the vendor Taconic [24] and Wang 2012 [25], C57BL/6N mice are expected to express at least diet-induced-obesity (DIO) and pre-diabetic conditions at 18 weeks of age when fed HF diet from week five. Hence, the timeframe of 13 weeks was chosen to ensure a clear effect of diet. All mice were sacrificed by cervical dislocation, and fecal content from the mice cecum and colon was sampled and suspended in 800 µL autoclaved 1× PBS (NaCl 137 mM, KCl 2.7 mM, Na_2_HPO_4_ 10 mM, KH_2_PO_4_ 1.8 mM). All samples were immediately stored at −80 °C.

### 2.2. Pre-Processing of Fecal Samples Prior Virome and Total DNA Extraction

Cecum and colon samples were thawed and 300 µL of suspended fecal content was mixed with 29 mL autoclaved 1× SM buffer (100 mM NaCl, 8 mM MgSO_4_·7H_2_O, 50 mM Tris-HCl with pH 7.5), followed by homogenization in BagPage+® 100 mL filter bags (Interscience, Saint-Nom-la-Bretèche, France) with a laboratory blender (Stomacher 80, Seward, UK) at medium speed for 120 seconds. The filtered homogenized suspension was subsequently centrifuged using an Allegra^TM^ 25R centrifuge (Beckman Coulter, Brea, CA, USA) at 5000× *g* for 30 min at 4 °C. The fecal supernatant was sampled for viral DNA extraction and the fecal pellet was re-suspended in 1× SM-buffer for bacterial DNA extraction. All laboratory procedures were performed aseptically and with BioSphere® filter tips to avoid contamination.

### 2.3. Bacterial DNA Extraction, Sequencing, and Pre-Processing of Raw Data

Tag-encoded 16S rRNA gene amplicon sequencing was performed on a Illumina NextSeq using v2 MID output 2 × 150 cycles chemistry (Illumina, San Diego, CA, USA). DNA extraction and library building for amplicon sequencing were performed in accordance with Krych et al. [26]. The average sequencing depth (Accession: PRJEB32231, available at ENA) for the cecum 16S rRNA gene amplicons was 318,395 reads (min. 47,182 reads and max. 808,971 reads) and 168,388 reads for colon (min. 47,004 reads and max. 223,787 reads), see Appendix A for further details. The raw NextSeq generated dataset containing pair-ended reads, with corresponding quality scores, were merged and trimmed using fastq_mergepairs and fastq_filter scripts implemented in the UPARSE pipeline [27]. The minimum overlap length of trimmed reads was set to 100 bp. The minimum length of merged reads was 130 bp. The max expected error E = 2.0, and first truncating position with quality score N ≤ 4. Purging the dataset from chimeric reads and constructing de novo zero-radius operational taxonomic units (zOTUs) were conducted using the UNOISE pipeline [28]. The k-mer based SINTAX [29] algorithm was used to predict taxonomy using the Ribosomal Database Project (Release 11, update 5) [30] as well as Greengenes (v13.8) [31] 16S rRNA gene collection as a reference database. The zOTU’s will subsequently be referred to as bacterial OTU’s (bOTU’s) to differentiate from the viral counterpart. Bacterial density in the cecum and colon content was estimated by quantitative real-time polymerase chain reaction (qPCR) as previously described [23], using 16S rRNA gene primers (V3 region) as applied for the amplicon sequencing [26]. Standard curves were based on total DNA extracted from *Escherichia coli* K-12 containing seven copies of the 16S rRNA gene.

### 2.4. Viral DNA Extraction and Sequencing

The fecal supernatant from the pre-processing was filtered through 0.45 µm Minisart® High Flow PES syringe filter (Cat. No. 16533, Sartorius, Göttingen, Germany) to remove bacteria and other larger particles for optimally maintaining the viral community [32,33,34]. The filtrate was concentrated using Centriprep® Ultracel® YM-50K units (Cat. No. 4310, Millipore, Burlington, MA, USA), which consist of an inner and outer tube. The permeate in the inner tube was discarded several times during centrifugation with 1500× *g* at 25 °C until approximately 500 µL was left in the outer tube. This was defined as the concentrated virome. The 50 kDa filter from the Centriprep® was removed by a sterile scalpel and added to the concentrated virome and stored at 4 °C until DNA extraction. 140 µL of virome was treated with 2.5 units of Pierce™ Universal Nuclease (Cat. No. 88701, ThermoFisher Scientific, Waltham, MA, USA) for 3 min. prior to viral DNA extraction to remove free DNA/RNA molecules. Based on the NetoVIR protocol [35], the nucleases were inactivated by 560 µL AVL buffer from the QIAamp® Viral RNA Mini kit (Cat. No. 52904, Qiagen, Hilden, Germany) used for viral DNA extraction. The NetoVIR protocol was followed from step 11–27, however, the AVE elution buffer volume was adjusted to 30 µL. The extracted viral DNA was stored at −80 °C prior to viral genome amplification. The Illustra Ready-To-Go GenomiPhi V3 DNA Amplification Kit (Cat. No. 25-6601-96, GE Healthcare Life Sciences, Marlborough, MA, USA) was used for genome amplification (expected avg. size of 10 kbp) of both dsDNA and ssDNA viruses for downstream analysis, whereas RNA viruses were not amplified. The instructions of the manufacturer were followed. However, the DNA amplification was changed to 30 min., instead of 90 min., to decrease the bias of preferential amplification of ssDNA viruses [36,37,38]. Genomic DNA Clean and Concentrator™-10 units (Cat. No D4011, Zymo Research, Irvine, CA, USA) were used to remove DNA molecules below 2 kb according to the instructions of the manufacturer. Prior library construction, the DNA concentrations of the clean products were measured by Qubit HS Assay Kit (Cat. No. Q32854, Invitrogen, Carlsbad, CA, USA) using a Varioskan Flash 3001 (Thermo Scientific, USA). Viral DNA libraries were generated by Nextera XT DNA Library Preparation Kit (Cat. No. FC-131-1096, Illumina, USA) by a slightly modified manufactures protocol divided into “Genomic DNA tagmentation” and “PCR clean-up”. Genomic DNA tagmentation: 5 µL Tagment DNA Buffer, 2.5 µL genomic DNA (in total 0.5 ng DNA), 2.5 µL Amplicon Tagment Mix, incubated at 55 °C for 5 min. followed by hold on 10 °C where 2.5 µL Neutralize Tagment Buffer was added and incubated at room temperature (RT) for 5 min. Then 7.5 µL Nextera PCR Mix and 2.5 µL of each Nextera Index primers i5 and i7 were added to a total volume of 25 µL and followed by PCR on SureCycler 8800. Cycling conditions applied were: 72 °C for 3 min., 95 °C for 30 s, 16 cycles of 95 °C for 30 s, 55 °C for 30 s, and 72 °C for 30 s, followed by final step at 72 °C for 5 min. PCR clean-up: 25 µL PCR product was mixed with AMPure XP beads (Beckman Coulter Genomic, USA), and incubated for 5 min. at RT and mounted to the magnetic stand for 2 min before continuing. The supernatant was removed, and each sample was washed with 150 µL of 80% ethanol twice. 27 µL of PCR-grade water was added, incubated at RT for 2 min., and mounted to a magnetic stand for 2 min. before sampling of 25 µL clean DNA products. The average sequencing depth (Accession: PRJEB32231, available at ENA) for the cecum viral metagenome was 829,533 reads (min. 212,545 reads and max. 1,621,360 reads) and 456,452 reads for colon (min. 63,183 reads and max. 643,913 reads), see Appendix A for further details.

### 2.5. Processing of Metagenome Sequencing of VLPs and Sequence-Based Knowledge

The raw reads were trimmed from adaptors and barcodes using Trimmomatic v0.35 (>97% quality [39] [seedMismatches: 2, palindromeClipThreshold: 30, simpleClipThreshold:10, LEADING: 15, MINLEN: 50], removed from ΦX174-control DNA and de-replicated (Usearch v10) [27]. Non-redundant/high-quality reads with a minimum size of 50 nt were retained for viromes reconstructions and downstream analyses. As quality control, the presence of non-viral DNA was quantified using 50,000 random forward-reads from each sample, which were queried against the human genome, as well as all the bacterial, viral, plant and mouse genomes hosted at NCBI using Kraken2 [40]. Similarly, reads were blasted against the non-redundant protein database available at UniProtKB/Swiss-Prot (-evalue 10^−3^, -query_cov 0.6, -id 0.7), the ribosomal 16S rRNA (GreenGenes v13.5 [31]) and 18S rRNA (Silva, release 126 [41]) databases (-evalue 10^−3^, -query_cov 0.97, -id 0.97). For each sample, reads were subjected to within-sample de novo assembly. For each sample, an assembly was carried out using Spades v3.5.0 [42,43] [using paired and unpaired reads] and the scaffolds (here termed “contigs”) with a minimum length of 1000 nt were retained. Contigs generated from all samples were pooled and de-replicated by multiple blasting and removing those contained in over 90% of the length of another (90% similarity) contig, as outlined by Reyes et al. [44]. To check the presence of non-viral DNA contigs, de-replicated contigs were evaluated according to their match to a wide range of viral proteins, [viral non-redundant RefSeq, virus orthologous proteins (www.vogdb.org), and the prophage/virus database at PHASTER (www.phaster.ca [45])], reference independent k-mer signatures [VirFinder [46]], viral genomes RefSeq [Kraken2] as well as their match to bacterial [NCBI, Kraken2 (--confidence 0.08)], plant [NCBI, Kraken2 (--confidence 0.3)], mouse [NCBI, Kraken2 (--confidence 0.1)] and human genomes [NCBI, Kraken2 (--confidence 0.1)]. All contigs matching viral proteins, viral k-mers, including those that did not match any database, were subsequently retained and categorized as viral-contigs.

### 2.6. Viral-Operational Taxonomic Unit (vOTU) Designation

Following assembly and quality control, high-quality/dereplicated reads from all samples were merged and recruited against all the assembled contigs at 95% similarity using Subread [47] and a contingency-table of reads per Kb of contig sequence per million reads sample (RPKM) was generated with a 10× coverage threshold, here defined as the vOTU-table (viral-operational taxonomic unit). Taxonomy of contigs was determined by querying (USEARCH-ublast, e-value 10^−3^) the viral contigs against a database containing taxon signature genes for virus orthologous group hosted at www.vogdb.org.

### 2.7. Bioinformatic Analysis of Bacterial and Viral Communities

Prior any analysis the raw read counts in the vOTU-tables were normalized by reads per kilobase per million mapped reads (RPKM) [48], since the size of the viral contigs is highly variable [49]. OTU’s which persisted in less than 5% of the samples were discarded to reduce noise, however, still maintaining an average total abundance close to 98%. Cumulative sum scaling [50] (CSS) was applied for analysis of β-diversity to counteract that a few bacterial and vOTU’s represented larger count values, and since CSS has been benchmarked with high accuracy for the applied metrics (Bray–Curtis, Sørensen-Dice, weighted-UniFrac, unweighted-Unifrac) [51]. CSS normalization was executed using the Quantitative Insight Into Microbial Ecology 1.9.1 [52] (QIIME 1.9.1) normalize_table.py, an open source software package for Oracle Virtual Box (Version 5.2.26). The viral and bacterial α-diversity analysis was based on, respectively, RPKM normalized and raw read counts to avoid bias with rarefaction [53]. This was supported by a comparison of the bacterial α-diversity (Shannon index) based on both the raw read counts and the rarefied read counts, see Appendix A. QIIME 2 (2019.1 build 1548866877) [52] plugins were used for subsequent analysis steps of α- and β-diversity statistics. Weighted (w) and unweighted (u) UniFrac [54] dissimilarity metrics represented the bacterial phylogenetic β-diversity analysis, whereas the non-phylogenetic β-diversity analysis was done by Bray–Curtis dissimilarity and Sørensen–Dice. The Shannon, Simpson and Richness indices represented likewise the determined α-diversity measures. The R-scripts A-diversity.R, Taxonomic-Binning.R, and Serial-Group-Comparison.R from the RHEA [55] pipeline (version 1.1.1.) were applied to detect taxonomy differences between groups with a relative abundance threshold at 0.25%. PERMANOVA test was performed with the function adonis() from the vegan package for R [56]. Wilcoxon rank sum test evaluated pairwise taxonomic differences, whereas ANOSIM and Kruskal–Wallis were used to evaluate multiple group comparisons. Venn diagrams were obtained with the web platform MetaCoMET [57], where bacterial and vOTU’s with less than 100 reads in any sample were discarded, and shared OTU’s were defined as present in 80% of the samples within a group (persistence). BLASTX v. 2.7.1 [58] database was applied to annotate and evaluate the presence of known integrase genes with a minimum *E*-value = 10^−3^ and alignment length at 51 bp amongst vOTU’s with a contig size above 3000 bp.

## 3. Results

Male C57BL/6N mice (*n* = 54) were purchased from Taconic (TAC), Charles River (CR), and Janvier (JAN). One-third of the mice were sacrificed at arrival as baseline, while the remaining were fed either a low-fat (LF) diet or a high-fat (HF) diet for 13 weeks until endpoint. Cecum and colon content were sampled from each individual mouse after being sacrificed. Here only results describing cecum samples will be reported. Complete equivalent analysis of colon samples can be found in Appendix A. The bacterial density in cecum samples was determined by qPCR to an average of 1.52 × 10^10^–3.79 × 10^10^ 16S rRNA gene copies/g. A t-test showed a significantly lower count of 16S rRNA gene copies/g when comparing HF vs. LF (*p* = 0.0017) and HF vs. baseline (*p* = 0.0008), and a significant difference (*p* < 0.0106) between the three vendors on LF diet, see Appendix A. For the viral community a total of 25019 contigs were assembled, of which 8503 contigs were characterized as viral origin and additional 7010 contigs as unclassified DNA (total 15513 contigs). The viral and unclassified DNA represented the initial vOTU-table, and 74.27% of the total Illumina NextSeq reads (Appendix A). The residual contigs (total 9506 contigs) were identified as bacteria, human, mouse, and plant DNA and represented 25.73% of the total Illumina NextSeq reads. More than 37% of the vOTU’s were taxonomically assigned to known viral orders or families. 

### 3.1. Gut Microbiota Diversity and Composition of C57BL/6N Mice from Three Vendors

Based on statistical differences of a Kruskal–Wallis group analysis, the effect of vendor (H = 14.4, *p* = 0.0007) on the bacterial Shannon diversity index exceeded the effect of the diet, as the latter had no significant (H = 0.48, *p* = 0.488) impact (Figure 2a,c). The Shannon index of the viral community was affected significantly by both vendor and diet (*p* < 0.02) (Figure 2b,c). Only minor variations in the statistical significance (Figure 2c) showed that the effect of vendor on the Shannon index seemed maintained from baseline to endpoint for both the bacterial and viral community. When comparing with the baseline, the bacterial and viral Shannon index of all three vendors decreased significantly (*p* < 0.025, see Appendix A for a statistically pairwise comparison of all groups) after the mice were fed HF or LF diet, except for the viral community of JAN LF, Figure 2b. Similar tendencies were observed with other α-diversity indices (Appendix A) and for an equivalent analysis of the colon samples (Appendix A), along with a statistically pairwise comparison of all groups (Appendix A). The top 10 most abundant vOTU’s (viral contigs) represented from 65.2% to 93.9% (median at 81.8%) of the relative abundance in each group, see Appendix A.

Analysis of similarities (ANOSIM) showed that vendor strongly influenced both gut bacterial and viral composition (Figure 3c). Additionally, diet had a significant effect, though not as pronounced as the vendor effect (as illustrated by the lower R-values, Figure 3c). A PERMANOVA test supported that the vendor effect exceeded the effect of diet on the β-diversity (Appendix A). The viral and bacterial community of all vendors and diets at baseline and endpoint were pairwise significantly (*p* < 0.007) separated (R > 0.652), see Appendix A. The bacterial (Figure 3a) and viral (Figure 3b) composition developed in similar directions from baseline to endpoint, however, the unique composition, which originated from the vendor maintained the significant separation observed at baseline (Figure 3c). Similar results were observed for the colon microbiota (Appendix A), and regardless of the β-diversity metric applied for the analysis, see Appendix A.

### 3.2. Taxonomic Abundance of the Bacterial and Viral Components

Several abundant bacterial genera significantly (*p* < 0.05) differed between the three vendors (Appendix A). Amongst these, especially *Prevotella* spp., *Alistipes* spp., *Desulfovibrio* spp., and *Akkermansia muciniphila* stood out, see Figure 4c–f. TAC mice had almost no *A. muciniphila* (Figure 4f)*, Prevotella* spp. (Figure 4c)*,* or *Alistipes* spp. (Figure 4d) whereas JAN mice had higher abundance of *Desulfovibrio* spp. (Figure 4e) compared to both CR and TAC. The viral community was clearly dominated by the family *Microviridae* whereas the order Caudovirales, *Siphoviridae, Podoviridae, Myoviridae,* and unclassified viruses constituted the remainder (Figure 4g). The vendors on LF diet had significantly (*p* < 0.05) fewer *Microviridae* (Figure 4a) compared to the HF diet, and opposite for Caudovirales (Figure 4b) expect TAC. See Appendix A for the bacterial and viral taxonomic binning of individual samples, and Appendix A for equivalent analysis of the colon microbiota.

### 3.3. Shared Taxonomies of Viral and Bacterial Entities amongst Three Vendors

Venn diagrams were made to follow the development of shared b- and vOTUs between the three vendors from baseline to endpoint, see respectively Figure 5a,b for the bacterial and viral community. The sum of the relative abundance represented by shared b- and vOTUs is illustrated by bar charts in Figure 5c and the associated taxonomies are listed in Appendix A. Only a few vOTU’s were shared between mice from the different vendors. At baseline only one vOTU (Figure 5b) was shared between all three vendors and constituted less than 0.05% of the viral abundance (Figure 5c). In contrary, after the dietary intervention (endpoint) the LF and HF vendor groups shared, respectively, 21 and 18 vOTU’s (Figure 5b) representing more than 60% of the relative viral abundance (Figure 5c). The shared vOTU’s only represented *Microviridae,* Caudovirales, and unclassified viruses. Independently of diet and time, a “core-virome” of all eighteen mice from each vendor was represented by respectively 20, 16, and 10 vOTUs for Taconic, Charles River, and Janvier, see Appendix A. The majority of these “core” vOTUs were shared between Taconic and Charles River, whereas all 10 “core” vOTUs from Janvier were unique, thus indicating the possibility of a vendor-specific “core virome” for C57BL/6N mice purchased at Janvier. Changes in the relative abundance of the shared bOTU’s from baseline to the endpoint were less pronounced (Figure 5c). However, the number of shared bOTUs increased from baseline at 73 bOTUs to, respectively, 326 and 249 at endpoint for LF and HF diet (Figure 5a). Equivalent analysis of the colon microbiota can be found in Appendix A.

## 4. Discussion

Here we investigate the impact of vendor and diet (low-fat vs. high-fat) on the bacterial and viral community of C57BL/6N (B6) mice purchased from Taconic (TAC), Charles River (CR), and Janvier (JAN). Overall, we observed that the bacterial and viral community was diet-dependent, which is consistent with former studies [22,59]. As expected, the bacterial and viral composition were affected by both vendor and diet, but surprisingly, the effect of vendor clearly exceeded the effect of the diet. Mice from all three vendors followed the same developmental direction in composition from baseline until endpoint and maintained the separation between vendors for 13 weeks. Duerkop et al. 2012 [60] showed that phages harbored by the human commensal bacteria *Enterococcus faecalis* V583 play a key role in establishing and maintaining its dominance. Similar mechanisms of active prophages harbored in the bacteria that initially colonized the new-born mice, at each vendor facility, could at least partly explain the strong effect of vendor on the microbial composition observed in our study. Furthermore, it would be expected that most bacteria contain one or more prophages in their chromosome [61] and that lysogeny is prevalent in the gut microbiota [62] thereby influencing the gut virome. However, stochastic mechanisms, such as community shuffling, and killing/invasion of relatives [63], would be expected to show dynamic changes of the gut virome through time. In addition, the methods applied for virome extraction are optimized to contain free phage particles (virus-like particles) in the lytic phase regardless if it is a temperate or purely lytic phage. This is further supported by only 5.3% (22 of 418 vOTUs) of the vOTUs above 3000 bp containing prophage marker genes, such as integrases, indicating that a significant proportion of the gut virome represents lytic phages. The vOTUs containing integrase genes mainly belonged to the family *Microviridae* and order Caudovirales in accordance to the phage taxonomy composition showed in Figure 4g. The clear vendor-related separation observed on the β-diversity was maintained despite the removal of vOTUs containing integrase genes, see Appendix A. Moreover, there was a significant vendor effect on the bacterial and viral α-diversity, while the diet had no effect on the bacterial α-diversity. vOTU’s belonging to the family *Microviridae* constituted minimum 60% of the relative viral abundance, whereas the order of Caudovirales and unclassified viruses represented the rest. Other studies support that *Microviridae* and Caudovirales are the main components in the human and animal virome [11]. The application of multiple displacement amplification (MDA) favours ssDNA viruses [36,37,38] like *Microviridae*, hence this might have influenced the relative abundance of *Microviridae*, however, MDA was shortened to 30 minutes to minimize this effect. The sequenced viral DNA libraries in this study did not include cDNA of RNA viruses since it was expected that the majority of phages have DNA genomes, although highly divergent RNA phages [64] could play important roles as well. Furthermore, it should be emphasized that the fraction of unclassified viruses might encompass Caudovirales or *Microviridae* phages that are not yet characterized. Only 10 vOTU’s constituted the majority (65–93%) of the total relative abundance, Appendix A. The relative abundance of the vOTU’s shared between the vendors clearly increased after being housed under the same conditions and diets, when compared to the shared vOTU’s at baseline, Figure 5c. As previously shown [16], the bacterial community of mice from three vendors clustered separately and differed in the relative abundance of important gut bacteria. We observed clear differences in the abundances of *Akkermansia muciniphila, Desulfovibrio* spp. and *Alistipes* spp. between vendors. *A. muciniphila, Prevotella* spp., and *Alistipes* spp. were almost absent in mice purchased from TAC and remained so even after 13 weeks of LF or HF diet. *A. muciniphila,* the only member of the genus in mice, has a strong influence on mucosal immune responses [65], and has been found to be inversely correlated to the incidence of type-1-diabetes in NOD mice [66]. *A. muciniphila* may also offer some protection against type-2-diabetes in diet-induced obese (DIO) mice [67], while it seems to be positively correlated to the development of colon cancer in azoxymethane-induced mice [68]. *Desulfovibrio* spp. are positively correlated with low-grade inflammation and obesity [69]. *Alistipes* spp. strongly influences metabolic profiles in feces of mice [70] and in a mouse model of autism a high level of *Alistipes* spp. in the gut correlated to a low level of serotonin in the ileum [71]. Stress-induced by housing mice on grid floors increases the abundance of *Alistipes* spp. [72]. *Prevotella copri* may increase the severity of dextran sulfate sodium (DSS)-induced colitis in mice [73], and the protective effects of *Caspase-3* knockout in mice may be counteracted by co-housing with wild type mice because these transfer *Prevotella* spp. to the knockout mice [74].

Howe et al. 2016 suggest that dietary history could have a distinct impact on the viral functional profile [3], and Ericsson et al. 2015 investigated the effect of the genetic background of mice on the fecal microbiota [75]. In addition to the effect of vendor [21] and diet [76] on the murine gut microbiota, mouse genetics have also been found to contribute to the variance in the murine gut microbiota [77]. Furthermore, the reproducibility of experiments is challenged by variations in housing conditions [75,78]. Thus, variation in the handling at the vendor housing facilities might explain the difference in the GM profiles even though the mice were of the same B6 strain. So, there are good reasons to assume that mice models based on mice from each of the three vendors, at least in some cases, will show phenotypic differences as well. In conclusion, to the best of our knowledge, this is the first study highlighting significant differences in the gut viral community of C57BL/6N mice from different vendors. It shows that vendor has pronounced effect on not only the gut bacterial community but also the gut virome, which has profound implications for future studies on the impact of the gut virome on GM interactions and host health.

## Figures and Tables

**Figure 1 viruses-11-00435-f001:**
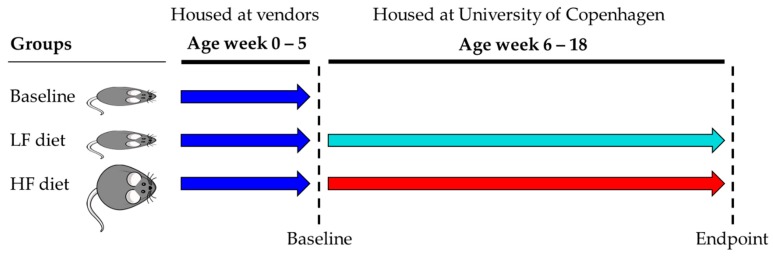
C57BL/6N mice were purchased from three different vendors: Taconic (*n* = 18), Charles River (*n* = 18), and Janvier (*n* = 18) and each vendor were divided into baseline, low-fat (LF) diet, and high-fat (HF) diet groups. The baseline mice (*n* = 6 pr. vendor) were sacrificed at arrival (age week five) and the LF and HF diet groups (*n* = 12 pr. vendor) were fed for 13 weeks before being sacrificed at endpoint (age week 18). Blue arrow = baseline, cyan arrow = LF diet, red arrow = HF diet.

**Figure 2 viruses-11-00435-f002:**
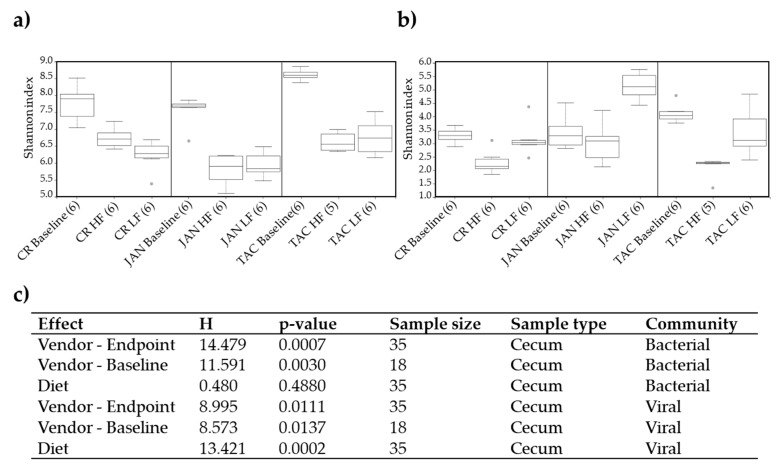
Shannon index of the caecal (**a**) bacterial and (**b**) viral community at baseline (five weeks of age) and after 13 weeks on low-fat or high-fat diet (18 weeks of age), respectively. The parentheses show the number of samples from each group included in the plot and grey dots indicate outliers. (**c**) Kruskal–Wallis group analysis of the Shannon diversity index of the effects of diet and vendor at baseline and endpoint. LF = low-fat diet, HF = high-fat diet, CR = Charles River, JAN = Janvier, TAC = Taconic.

**Figure 3 viruses-11-00435-f003:**
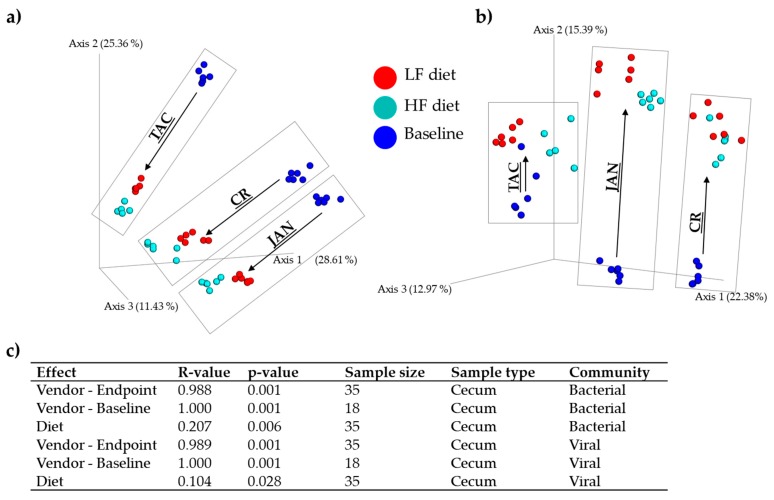
Bray–Curtis dissimilarity metric PCoA based plots of (**a**) the caecal bacterial community and (**b**) viral community at baseline (five weeks of age) and at endpoint after 13 weeks on low-fat or high-diet (18 weeks of age), respectively. (**c**) ANOSIM of the Bray–Curtis distances of the effects of diet and vendor at baseline and endpoint. Grey boxes frame the samples associated with the mice vendor. LF = low-fat diet, HF = high-fat diet, CR = Charles River, JAN = Janvier, TAC = Taconic.

**Figure 4 viruses-11-00435-f004:**
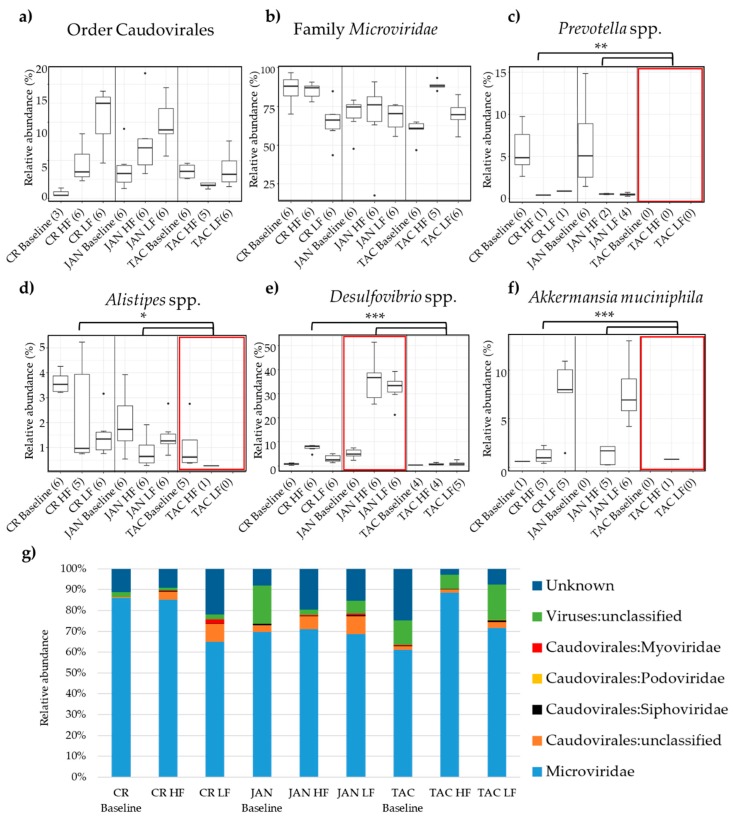
Relative abundance of (**a**) the order Caudovirales and (**b**) the family *Microviridae*. Differences in the relative abundance of *Prevotella* spp., *Alistipes* spp., *Desulfovibrio* spp., and *Akkermansia muciniphila* between vendors and diet are illustrated in respectively (**c**), (**d**), (**e**), and (**f**). The most abundant viral taxonomies are illustrated by bar plots in (**g**). The parentheses show the number of samples from each group included in the plot. Black dots indicate outliers and the red boxes mark the vendor with interesting differences in bacterial abundance. Black branches and stars mark the significant bacterial differences in abundance between vendors, * = *p* < 0.05, ** *p* < 0.005, *** *p* < 0.0005 based on pairwise Wilcoxon rank sum test. Abbreviations: LF = low-fat diet, HF = high-fat diet, CR = Charles River, JAN = Janvier, TAC = Taconic.

**Figure 5 viruses-11-00435-f005:**
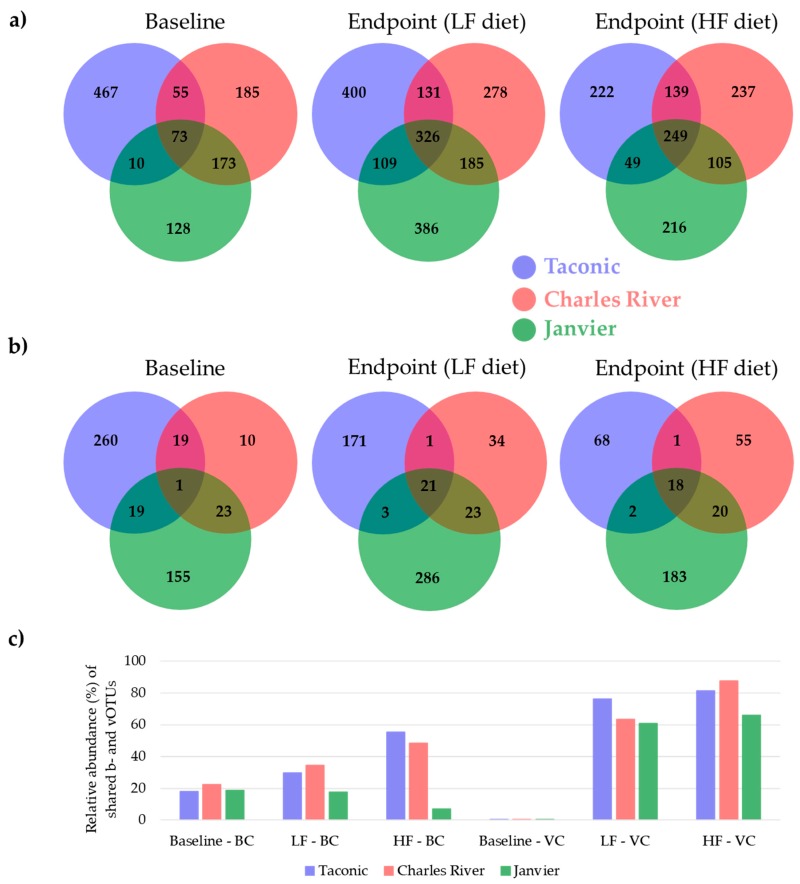
Venn diagrams illustrating the number of shared caecal (**a**) bacterial and (**b**) viral OTUs (b- and vOTU’s) amongst mice purchased from three vendors at baseline (five weeks old) and endpoint (18 weeks old) on either high-fat (HF) or low-fat (LF) diet. The numbers inside the Venn diagram indicate the amount of shared OTUs. (**c**) Bar plot illustrating the sum of the relative abundance of the shared b- and vOTUs from baseline to endpoint. BC = Bacterial community, VC = viral community.

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
