# Peer review of "Mouse Vendor Influence on the Bacterial and Viral Gut Composition Exceeds the Effect of Diet"

_viruses, 2019, doi:10.3390/v11050435_

Round 1
Reviewer 1 Report
In this manuscript, Rasmussen et al. sequence the composition of mouse intestinal bacteria and bacteriophage populations from different mouse vendors on different diets. They find that vendor differences are the main influence on microbiota compositions exceeding those of the changes induced by diet. While the manuscript has some interesting findings, the manuscript needs revision before warranting publication.
Major Concerns:
1. The choice of using five-week-old mice as a baseline is concerning. Microbiota changes would likely occur over a 13-week span regardless of changing diet and the environment due to puberty. The authors need to, at the very least, provide a rationale for the time-frame from “baseline to endpoint.” For example, was it entirely necessary for the mice to be on a diet for 13-weeks to see these effects?
2. It’s not surprising that bacteriophage populations vary since there is a difference in intestinal bacteria from each vendor and diet. Most bacteria harvest one or more phages in their chromosome, and the authors do not discuss the correlation between the two. For example, do the abundance of specific viral families correlate to prophage sequence in distinct bacterial families that are also abundant in these samples?
Minor Concerns:
1. The authors sequence bacteriophages based on DNA extraction. While I agree that the majority of bacteriophages have DNA genomes, some RNA bacteriophages exist and are missed by this analysis (Krishnamurthy et al. PLoS Biol. 2016). The authors should at least address this in the discussion.
2. The figures are not adequately described in the text. For example, the authors discuss portions of the figure without detailing it to the reader. The authors should revise the text to be clear which panel of the figure they are talking about (i.e., Fig 1A or Fig 1B).
3. The box and whisker plots provided in the manuscript are ignored in the text. Additionally, the asterisks (or circles) in these, which I believe are potential outliers, are not described at all. These need to be clarified because to an average reader it looks like the authors are trying to show significance between two groups.
4. Bacteriophages are known to alter the intestinal commensal bacteria (Duerkop et al. PNAS 2012). These data should be discussed in relation to the authors’ findings.
Author Response
Dear Reviewer
Thank you for improving our manuscript with your constructive comments. Please find our respond with red font in the attached Word-document.

Reviewer 2 Report
The study by Rasmussen et al. investigates an interesting problem of gut microbiome and virome variation in C57BL/6N mice procured from different vendors. The vendor effect seen in this study seems to be persistent and exceeding the effect of diet.
In my opinion, the study could greatly benefit if different mice genetic backgrounds were included and procured from the same set of vendors. That would enable the authors to compare relative effects of genetic background, inidividual mice variations, vendor and diet.
Other than that, the study is very well done and well presented. The methods section is very detailed. The choice of bioinformatical and biostatistical approaches is well justified and scientifically sound. Some improvement could be done to the plots to make them more aesthetically appealing.
Line 40. I don't think there is sufficient data available to date to conclude that phages are responsible for shaping bacterial microbiota in the gut. Some studies highlighted strong correlation between the virome and the "bacteriome" composition (https://www.ncbi.nlm.nih.gov/pubmed/30763537), while others demonstrated the ability of phages to co-engraft together with their hosts in the FMT (https://www.ncbi.nlm.nih.gov/pmc/articles/PMC6288847/). But in general, it is still an unsolved "chicken and egg" dilemma: whether phages are responsible for shaping microbial communities, or just passively reflect on their composition? To the best of my knowledge, no definitive experimental proof of either of the two hypotheses has been provided so far.
Line 78. It is unclear why such strict anaerobic techniques were used if microbiome analyses were done by sequencing only?
Line 119. This is perhaps one of the most important factors affecting gut virome analysis (see https://www.ncbi.nlm.nih.gov/pubmed/26577924, https://www.ncbi.nlm.nih.gov/pubmed/25608871, https://www.nature.com/articles/srep16532, https://www.ncbi.nlm.nih.gov/pubmed/29631623).
Line 207. I think it would make sense to talk a little bit about the structure of viromes recovered by your sequencing pipeline before going into group vs. group comparisons. What fraction of reads could be aligned to your vOTU catalogue? What fraction of contigs could be taxonomically assigned and what major taxonomic groups wer detected? What was the outcome of your contamination checks performed as described in lines 159-164 and 169-175. I wonder how efficient was the removal of possible bacterial contamination.
Lines 234-241. I would suggest to do something like PERMANOVA test (e.g. adonis() from the 'vegan' package for R) in order to highlight the relative weight of factors (R2 for vendor vs. diet vs. residuals) on the microbiota composition. To me it would be more informative than comparing the ANOSIM R-values.
Line 280. Just wanted to confirm that the numbers in Venn digrams (panel B) relate to the number of shared individual vOTUs, not shared taxonomies. If that's the case, how many vOTUs were shared between different animals within each vendor? Was there such a thing as a vendor-specific "core virome"? From looking at panels A and B, it looks that microbiomes and viromes from different vendors converge over time, in terms of number of shared OTUs. Is that correct? Also, it would be good to know how abundant were those shared OTUs. I wonder, if it would be possible to replace Venn diagrams with barcharts, highlighting relative abundances of shared OTUs (maybe by using matching colours) in a broader context of vendor-specific microbiota and virome.
Author Response

(The authors gave the same response as above.)

Round 2
Reviewer 2 Report
I'd like to thank the authors for providing a detailed response to all my queries and suggestions. I have nothing more to add at this point. In my opinion the manuscript can be published in its present form.